# In-Depth Characterization of a Re-Engineered Cholera Toxin Manufacturing Process Using Growth-Decoupled Production in *Escherichia coli*

**DOI:** 10.3390/toxins14060396

**Published:** 2022-06-08

**Authors:** Natalia Danielewicz, Wenyue Dai, Francesca Rosato, Michael E. Webb, Gerald Striedner, Winfried Römer, W. Bruce Turnbull, Juergen Mairhofer

**Affiliations:** 1enGenes Biotech GmbH, Mooslackengasse 17, 1190 Vienna, Austria; natalia.danielewicz@engenes.cc; 2Department of Biotechnology, University of Natural Resources and Life Sciences, Muthgasse 11, 1190 Vienna, Austria; gerald.striedner@boku.ac.at; 3School of Chemistry and Astbury Centre for Structural Molecular Biology, University of Leeds, Leeds LS2 9JT, UK; w.dai@leeds.ac.uk (W.D.); m.e.webb@leeds.ac.uk (M.E.W.); w.b.turnbull@leeds.ac.uk (W.B.T.); 4Faculty of Biology, University of Freiburg, Schänzlestraße 1, 79104 Freiburg, Germany; francesca.rosato@bioss.uni-freiburg.de (F.R.); winfried.roemer@bioss.uni-freiburg.de (W.R.); 5Signaling Research Centers BIOSS and CIBSS, University of Freiburg, Schänzlestraße 18, 79104 Freiburg, Germany; 6Freiburg Institute for Advanced Studies (FRIAS), University of Freiburg, 79104 Freiburg, Germany

**Keywords:** cholera toxin, holotoxin-like chimaera, neuronal tracer, T2SS machinery, GM1 receptor, retrograde transport, GspD, fed-batch process, affinity chromatography

## Abstract

Non-toxic derivatives of the cholera toxin are extensively used in neuroscience, as neuronal tracers to reveal the location of cells in the central nervous system. They are, also, being developed as vaccine components and drug-delivery vehicles. Production of cholera-toxin derivatives is often non-reproducible; the quality and quantity require extensive fine-tuning to produce them in lab-scale settings. In our studies, we seek a resolution to this problem, by expanding the molecular toolbox of the *Escherichia coli* expression system with suitable production, purification, and offline analytics, to critically assess the quality of a probe or drug delivery, based on a non-toxic derivative of the cholera toxin. We present a re-engineered Cholera Toxin Complex (rCTC), wherein its toxic A1 domain was replaced with Maltose Binding Protein (MBP), as a model for an rCTC-based targeted-delivery vehicle. Here, we were able to improve the rCTC production by 11-fold (168 mg/L vs. 15 mg/L), in comparison to a host/vector combination that has been previously used (BL21(DE3) pTRBAB5-G1S). This 11-fold increase in the rCTC production capability was achieved by (1) substantial vector backbone modifications, (2) using *Escherichia coli* strains capable of growth-decoupling (V strains), (3) implementing a well-tuned fed-batch production protocol at a 1 L scale, and (4) testing the stability of the purified product. By an in-depth characterization of the production process, we revealed that secretion of rCTC across the *E. coli* Outer Membrane (OM) is processed by the Type II secretion-system general secretory pathway (gsp-operon) and that cholera toxin B-pentamerization is, likely, the rate-limiting step in complex formation. Upon successful manufacturing, we have validated the biological activity of rCTC, by measuring its binding affinity to its carbohydrate receptor GM1 oligosaccharide (K_d_ = 40 nM), or binding to Jurkat cells (93 pM) and delivering the cargo (MBP) in a retrograde fashion to the cell.

## 1. Introduction

Native cholera toxin (CTx), produced by the host *Vibrio cholerae* (*V. cholerae*), belongs to the AB_5_ family toxins and is composed of a catalytic A domain (CTxA, separated into A1 and A2 subunits) anchored in rings of five identical B subunits (CTxB) [1,2,3]. Subunit A1 catalyzes the ADP-ribosylation of α-subunits of G protein (Gsα), a GTP-binding regulatory protein, to activate the enzyme adenylate cyclase. This leads to overproduction and intracellular accumulation of the second messenger 3′,5′-cyclic AMP (cAMP), from ATP. Raised cAMP levels result in the hypersecretion of chloride and bicarbonate into the intestinal lumen, leading to a rapid loss of water from the intestine and resulting in the characteristic cholera stool. The A1 domain is the only toxic element of the complex and is covalently bound to the non-toxic A2 helix, which tethers A1 to the pentameric CTxB. CTxB recognizes the GM1 ganglioside on the surface of target cells and acts as a carbohydrate-binding protein, leading to endocytosis and retrograde trafficking of the entire complex, from the cell surface to the Golgi and endoplasmic reticulum, from where the A1 domain is released into the cytosol. GM1 is found in relatively high abundance in the nervous system, e.g., the periphery of motor neurons at the neuromuscular junction [4]. Injection of CTxB into a muscle can, thus, allow uptake of the protein into motor neurons for retrograde trafficking to the cell body, which is located in the spinal cord or brainstem. Wild-type CTxB, re-engineered CTxB proteins, and their complexes have, therefore, been widely exploited as a neuronal tracer [5,6,7,8]. Non-toxic AB_5_ complexes of analogous bacterial toxins have, also, been reported for the targeted delivery of proteins to the cytosol of neurons [9]. CTxB is, also, a component of the Dukoral^®^ anti-cholera vaccine [10], and there has, also, been widespread interest in using AB_5_ bacterial toxins, and their B-subunits, as platforms for vaccine development [11,12,13,14,15].

The widespread interest and applications of bacterial toxin derivatives, as neuronal tracers and vaccine components, have to be complemented with a high-yield manufacturing process, to make these pharmaceutical approaches economically viable. The production of the CTxB component alone has been proven productive, achieving yields up to 1 g/L (Dukoral^®^ CTxB vaccine) [16,17]. A wide range of non-toxic AB_5_ recombinant cholera toxin complexes (rCTCs) have been reported [15,18,19], however, their production is much less efficient. While the A2 subunit can facilitate assembly of the pentamer [20] and helps to maintain holotoxin stability during the uptake and transport into cells [21,22], the A2 peptide is, to some extent, proteolytically digested during microbial production, leaving an overall lower yield of this holotoxin [23].

A prerequisite for obtaining high yields of A2-CTxB tracer is a detailed understanding of the native-expression process. During homologous (*V. cholerae*) and heterologous (*Escherichia coli*; *E. coli*) expression, the CTx is secreted via the type II secretion system (T2SS) [24], and CTxB possesses signal peptides, directed towards the Sec pathway, to be translocated across the inner membrane in unfolded form [25], however, the tat secretion pathway has, also, been used for rCTCs [19]. These subunits fold and spontaneously assemble into the holotoxin, to be exported extracellularly by the general secretory pathway, (gsp) operon [24,25,26]. The ratio of A- and B-subunits, before assembly, is 2:5 for the analogous *E. coli* heat-labile toxin [27]. Uncomplexed A-subunit remains in the periplasmic space, where it undergoes a slow degradation process. The AB_5_ holotoxin is exported across the Outer Membrane (OM) to the extracellular medium [3,28]. Structural analysis of the secretion complex involves a piston-driven secretion mechanism [26], in which correct interaction between CTxB and the periplasmic GspD vestibule is decisive for the holotoxin secretion to the extracellular space. The abovementioned arguments indicate that important conditions for extracellular recombinant holotoxin expression are: (i) signal peptides for periplasmic expression of holotoxin subunits; (ii) combination of transcription and translational fine-tuning to deliver a 2:5 ratio of subunits expressed to the periplasm; and (iii) sufficient expression of CTxB to export the holotoxin into the extracellular milieu. Concerning those conditions, we present optimized rCTC neuronal tracer production at a 1 L fed-batch scale. In this study, we used a non-toxic A2 subunit, with a Maltose Binding Protein (MBP) soluble tag, to enable high-level production of non-toxic rCTC [18]. Additionally, we present an in-depth characterization of rCTC stability, after purification, addressing its capacity to recognize GM1 receptors, and its potential for retrograde tracking in cultured cells.

## 2. Results and Discussion

### 2.1. rCTC Construct Design

Our rCTC construct design (Figure 1a) was similar to that previously reported by Jobling and Holmes [15], with a polycistronic construct comprising the A2 and CTxB genes, cloned into a pMALp5x plasmid. The LTIIb leader sequence was used for the CTxB gene, as it has been reported to give higher yields of CTx holotoxins in *E. coli* [29]. Periplasmic expression was used, as the formation of the CTxB disulfide bond is a prerequisite for pentamer assembly. The high-performance MBP-solubility tag [30], with its intrinsic ability to solubilize fusion partners [31,32,33], was intended to help maintain the A2 helix in solution, during the assembly of rCTC, but it, also, plays a role as a purification tag [34]. This amylose-specific tag, fused to the A2 helix, complements CTxB’s intrinsic ability to bind immobilized metal affinity chromatography (IMAC) resin and enables the two-step purification of the intact rCTC. Once the rCTC is expressed and purified, the MBP-tag can be removed by a highly sequence-specific cysteine protease from the Tobacco Etch Virus (TEV protease; Figure 1b), to allow further functionalization of the A2 peptide [18].

The toolbox of the rCTC construct was expanded for multiple growth-decoupling *E. coli* strains and plasmids suitable for high-yield expression (Table 1). Four different plasmids were matched with five different strains, resulting in eight variants (Figure 2). Each was given a specific ID, for better identification and labeling. Both V1 *ΔthyA* and B *ΔthyA* function without essential thymidine production, sustained by pET30a <MBP-A2_CTxB>ThyA_Cer plasmid. 

Original rCTC construct (pTRBAB5-G1S) [18], subcloned to pET30a<>Cer [35] that contains a Cer site to enhance plasmid stability [36], has increased promotor strength from tac (weak promotor) to T7 (strong promotor). The tac promotor regulated by RNA polymerase (RNAP) has to be replaced, before introducing the rCTC-bearing plasmid in the V (V1, V2, and V1 *ΔthyA*) strains [37,38]. Within the V strains, decoupling of protein production from cell growth is achieved by l-arabinose inducible expression of the *E. coli* RNA polymerase specific inhibitor peptide Gp2. The combination of the *E. coli* RNA polymerase-driven tac promotor and V strains would, then, be uncomplimentary for the rCTC expression, since the strains are intended to be used with the orthogonal T7 RNA polymerase and are not affected by the inhibitor Gp2.

For the pET30a<MBP-A2_CTxB>Cer plasmid, high production of rCTC was further optimized, by lowering the Ribosome Binding Site (RBS) strength of MBP-A2 from the original pMALp5x-derived construct (pTRBAB5-G1S), to reduce expression of MBP-A2 and maintain the ratio close to the 2:5 rCTC subunit ratio [27]. The RBS prediction was performed with the RBS Library calculator (https://salislab.net/, accessed on 22 May 2022). A single-point mutation introduced in pET30< MBP-A2_CTxB>Cer complements the original weaker promotor (tac) and reduces promotor strength to 0.09% (pET30a 0.09T7< MBP-A2_CTxB>Cer). This was established by introducing multiple single-point mutations at different base positions of the T7 promoter and examining the expressibility of the model protein, after induction (data not shown). 

An antibiotic-free plasmid-selection system was achieved with auxotrophic strains (V1 *ΔthyA* and B Δ*thyA*), where antibiotic-resistant genes were replaced with thymidine synthesize gene (*thyA*), suitable for ThyA autotrophs [39,40]. Such an auxotrophic strain eliminates toxicity brought by antibiotics and is a reliable selection system for large-scale rCTC production. Both V1 *ΔthyA* and B *ΔthyA* function without essential thymidine production, sustained by pET30a <MBP-A2_CTxB>ThyA_Cer plasmid.

### 2.2. rCTC Production Optimization

#### 2.2.1. Selection of Inducer Concentration for Large-Scale Production of rCTC 

Carbon-limited fed-batch-like high-throughput µ-bioreactor cultivations can provide adequate screening conditions for the identification of optimal induction parameters, delivering high-product yields. The µ-bioreactor system used, called Biolector^®^, allows for 48 parallel cultivations and online monitoring of Cell Dry Mass (CDM). Previously, optimized Biolector^®^ conditions for the B strain were used in our experimental set-up [41,42]. The goal of this screening work was to identify suitable isopropyl-β-thio-galactoside (IPTG) and arabinose (Ara) concentrations, for induction of the eight host-vector combinations producing rCTC. IPTG induces rCTC production in the B and V strains. Ara induces the expression of Gp2, the host RNAP inhibitor in V strains, but is, also, capable of inducing rCTC production via promotor cross-talk, as shown recently [35,38].

In these small-scale screening cultivations, we varied the amount of IPTG (0.01–1 mM) and arabinose (0.25–100 mM) added to the cultures, and we analyzed the influence of induction on cell growth, cell leakiness, and product formation. In Figure 3, induction experiments with two host-vector combinations, named V1 pET30a and V1 pET30a 0.09T7, are presented, as an example. The growth profiles define an Ara concentration of 1 mM, as sufficient to decouple cells from growth (Figure 3a). The reduction in CDM, caused by IPTG, is further presented by inducing cells with low Ara concentration (1 mM IPTG with 0.25 mM Ara) and suggests rCTC expression impacts the metabolic load of cell growth. In Figure 3b, we, also, observed clear growth-rate changes occurring during the batch and feed phases.

The End of Fermentation (EoF) samples were analyzed via flow-cytometry analysis using propidium-iodide (PI) staining, for testing membrane integrity and cell leakiness, during rCTC production (Figure 3c). Positive PI staining can then be an indicator for cell leakiness (after induction), if combined with extracellular DNA measurement (cell-lysis indicator). 

The shift of PI staining in a flow cytometry assay (Figure 3c) indicates an increase in transport through the microbial membrane, which increases the likelihood of rCTC extracellular expression, during a larger fermentation run (e.g., 1 L fed-batch cultivation). Permeability (Figure 3c) and productivity (Figure 3e) complement each other, in terms of accessing the best-performing induction condition. The PI staining method was further complemented with extracellular DNA concentration measurement, for differentiating cell leakiness from cell lysis. The extracellular DNA concentration did not increase at the end of fermentation (Figure 3e). This indicates low or no cell disruption, upon induction. PI staining of only induced cells suggests permeability, without complete disintegration of the cell membrane, during rCTC production. IPTG drives rCTC production to either increase periplasmic expression into the extracellular space or to maintain the intracellular expression, without completely disrupting cell-membrane integrity. 

An ELISA was implemented to determine the yield of the active, and soluble complex (Figure 3d,e). Ganglioside GM1 was used to coat the microtiter plate, to enable specific capture of the CTxB unit, while protein detection was performed using an anti-MBP antibody, to ensure that only intact rCTC was detected. Measuring the concentration of protein in a solution with ELISA is superior to the SDS-PAGE method (with standards of known concentration), which detects both active and non-active proteins, surrounded by a heavy background of Host Cell Proteins (HCPs). The ELISA allowed the detection of different quantities of rCTC, in response to changing the inducer concentration. In this small-scale experiment, the majority of rCTC remained inside the cells, with the highest expression achieved at induction, with 1 mM IPTG and 100 mM Ara. Even though rCTC is expected to be expressed, predominantly, extracellularly, it is often mentioned in the literature that this might not be the case in small-scale production [12,15,29]. Retention of rCTC in the periplasm is, likely, caused by delayed in-process production and accumulation of subunits (MBP-A2 and CTxB), at a concentration necessary for complex formation and, then, export to the extracellular milieu.

The same set of experiments was performed for all other variants (Appendix A). Each of the variants showed the highest productivity at full induction (1 mM IPTG with 100 mM Ara induction for V strains; 1mM IPTG for B strain); 1 mM IPTG 100 mM Ara induction was selected for V strains; and 1 mM IPTG for B strain inducers concentration was selected for 1 L scale rCTC production. 

#### 2.2.2. Understanding Factors That Influence rCTC Expression

Optimal expression of rCTC requires an in-depth understanding of transcriptional/translational machinery operating in the selected production strain. The expression levels of a selection of genes expected to be responsible for the extracellular expression of rCTC were measured with qPCR. In its native host, *V. cholerae*, the toxin is secreted using the general secretory pathway (eps or its analogue gsp in *E. coli*). In *E. coli* the gsp machinery is activated by the transcriptional activator SlyA [43,44,45,46,47] and negatively regulated by Fur and Hns, which act as transcriptional inhibitors that counter the effect of SlyA [46,47].

Here, we decided to assess the capabilities of B (wt BL21(DE3)) and V1 (mutant BL21 (DE3)) strains, carrying pET30a <MBP-A2_CTxB>Cer plasmid, to express rCTC product extracellularly. Extracellular expression of proteins can be determined for B strain mutants, by screening the expression level of GspD [48] from the gene cluster of the T2SS protein-secretion system. Based on these findings, we performed a simple shake-flask experiment and qPCR analysis, to determine the regulation of *gspD*, *gspE, gspG, slyA, fur*, and *mbp-a2* with *ctxB* genes (Figure 4). Since we expected no expression of *mbp-a2* and *ctxB* before induction (0 h), we used these values as an additional, internal control. DNA-binding transcriptional dual regulator *hns* was excluded from analysis, since it inhibits transcription initiation of *gspD* the same as *fur*.

V1 strain showed overall higher expression of rCTC and upregulation of *ctxB*, and *mbp-a2* agrees with upregulation of the OM protein, *gspD*. The rCTC is mainly upregulated after 6 h of induction (3.5-fold increase), which suggests that the production of the complex is expected to be later during the production process. This correlation of *ctxB*, *mbp-a2*, and *gspD* 6 h post-induction was not observed for the B strain. Therefore, we conclude that increased extracellular production observed for the V1 strain is due to increased activity of the T2SS later in production. Overall, the qPCR shows that extracellular expression of rCTC by the B strain and B strain mutants (e.g., the V1 strain) is possible, and that maximal expression levels are to be expected after a prolonged induction phase, where the GspD expression levels are sufficient for extracellular production. In contrary to the B strain, no upregulation for *slyA* and *fur* was observed in the V1 strain. Further testing is still required for determining the role of those genes in rCTC expression.

#### 2.2.3. Selection of Strain for rCTC Production

rCTC was expressed for 25 h (with 0.5 mM IPTG/100 mM Ara induction for V strains; 0.5 mM IPTG for B strain), purified by two-step affinity chromatography (IMAC and amylose column), and, then, concentrated by centrifugal ultrafiltration (Figure 5a). During the preliminary expression of rCTC in the 1 L scale fed-batch cultivation, we detected higher product titers present in the extracellular space, in comparison to the yields observed following small-scale fed-batch-like cultivation in Biolector^®^ (Figure 3 and Figure 5). Alongside those results, we detected a low amount of rCTC present intracellularly in the periplasm (data not shown). 

pO2, pH, Air, Temperature (Temp), Base Flow, Agitation, and Feed Flow parameters were measured during a fed-bath process in the Dasgip^®^ reactor (Figure 5b,c). Feed rate dictates the exponential or linear profile of fed-batch fermentation in the reactor. 

An analysis of the process parameters was implemented, to investigate the viability of cells during rCTC production. Determining the point at which rCTC production should be stopped, before toxicity, limits the viability of cells and has a direct consequence on the purification process. Lysing cells influence the overall viscosity of cell suspension, which results in a more demanding separation of media supernatants from the cell biomass. In Figure 5c, air input and extracellular DNA of the V1 pET30a 0.09T7 variant are increased after 25 h in the feed process (10 h in rCTC production). This indicates that a high concentration of rCTC causes cell lysis during the production process. End of Fermentation (EoF) concentration of rCTC produced by the V1 pET30a T70.09 variant surpassed other variants by up to 11-fold (Figure 5d), yet, the increased DNA [mg/g] content (15 h) suggests that there are benefits, upon earlier termination of the process. This rCTC production process has a substantially increased protein concentration (168 mg/L), with only moderately increased DNA concentration (267 µg/mL), but is, still, suitable for straightforward purification, avoiding possible high-viscosity issues (straightforward separation of biomass and filtration). The calculation performed, for comparing extracellular DNA concentration, was changed from mg/g CDM (Biolector^®^) to mg/L (Dasgip^®^). This is due to larger biological differences between the B strain and the CDM-decoupling V strain. Overall, the V1 pET30a 0.09T7 variant has improved the productivity of rCTC by 48-fold [11]. We observed an even higher increase in rCTC (168-fold), when comparing B pTRBAB5-G1S (shake flask) and V1 pET30a 0.09T7 (1 L fermenter). The selection of strain and plasmids, with fine-tuned fermentation conditions, solved problems for the low production yield of rCTC.

Determination of formulated rCTC concentrations was complemented with SDS-PAGE gels (Figure 6a and Appendix A). We observed that CTxB is produced later in the fermentation process (highest at 15–25 h), which is consistent with our qPCR data (Figure 4a). The band, at a size corresponding to MBP, begins to disappear at the point at which CTxB appears on gels (Figure 6a). This indicates that CTxB expression and consequent rCTC formation protect MBP-A2 from possible proteolytic digestion. The possibility that the disappearing MBP band originates from HCPs was excluded, by visualizing the content of extracellular aggregates solubilized with urea (Figure 6b). These findings were further confirmed with peptide mapping LC-ESI-MS/MS analysis, described in Appendix A. MBP detected in the SDS-PAGE originates from a plasmid, as it contains fragments of A2 or flexible linker (fX cleavage site), introduced by molecular cloning. HCPs present in large quantities suggest that the density of the SDS-PAGE band (Figure 6) does not necessarily correspond to high basal expression of MBP-A2, but, rather, a digested product with exposed charged residues. MBP-A2 with a modified solubility profile attracts charged HCPs, at a similar molecular weight (MW), into the extracellular milieu. The MBP-A2 cannot be exported extracellularly by T2SS machinery, without GspD-specific CTxB [27]. This suggests that either MBP can export helix with HCP carry-over, or *E. coli* has additional machinery that exports digested and potentially toxic proteins outside the cells. This phenomenon is observed in all variants and, so, is not strain specific.

Those results, also, suggest that initiation of earlier EoF for the V1 pET30a T70.09 variant will lead to substantially lower rCTC yields, due to delays in the process of CTxB production (Figure 6). It is then understood that effective rCTC production is around the threshold where rCTC is abundant, while not high enough to lead to complete cell lysis of the expressing strain.

#### 2.2.4. rCTC Inducer Titration in Selected Strains 1 L Scale

The best performing producers, V1 pET30a T70.09 and V1 Δ *thyA* pET30a variants, were selected for IPTG/Ara titration experiment in a 1 L scale fed-batch process (Figure 7). Based on preliminary induction optimization performed with the V1 strain in small-scale fermentation, we know that Ara induction (decoupling effect) can be performed at concentrations as low as 5 mM and IPTG induction (rCTC expression) as low as 0.01 mM [35]. rCTC-specific ELISA was performed, to determine the level of rCTC produced during the 25 h production process. We, also, compared the new data with the previous experiment, described in Section 2.2 (0.5 mM IPTG/100 mM Ara induction).

Figure 7 shows that V1 pET30a T70.09 requires the highest strength induction (0.5 mM IPTG/100 mM Ara), to maximize yield (219 mg/L), and V1 Δ *thyA* pET30a requires slightly lower IPTG induction (0.1 mM IPTG/100 mM Ara), for producing up to 358 mg/L. Based on the extracellular DNA concentration, air, O_2_, and base demands, observed during the course of fermentation, we suggest the V1 pET30a T70.09 variant partly lyse after 10 h induction. V1 Δ *thyA* pET30a gives the highest levels of expression, with optimal harvest between 5–15 h after induction. The best-performing strain after 2 is V1 pET30a T70.09, at the highest induction strength (0.5 mM IPTG/100 mM Ara), with a long production time that gives high yields of stable rCTC. The benefits of combining two variants into V1 Δ *thyA* pET30a T70.09 are yet to be researched.

Overall, the delivery of holotoxin into periplasm at a higher rate can affect the T2SS secretory machinery (GspD OM protein) [26]. Our data from the shake-flask experiment, using qPCR analysis, were further supported by transcriptomics, as GspD is upregulated at 1 h and 6 h, after induction for the empty V1 strain (Appendix A). The GspD expression increases even further, for a strain bearing the pET30a<MBP-A2_CTxB>Cer plasmid, and imposes greater demands on OM-protein-expression machinery, leading to cell lysis (Figure 7).

### 2.3. Stability and rCTC Quality Measurements

#### SEC-MALS and SEC Stability Testing

The volumetric yields (mg rCTC/L) and specific yields (mg rCTC/g CDM) at EoF were measured and compared between variants (Table 2). Purified and formulated rCTC was tested with SEC, SEC-MALS, and MS (Appendix A), to determine the complex stability after storage and its MW. No post-translational modifications were detected, and all variants had the same MW of ~107 kDa. The investigation concluded that, in addition to the rCTC peak and free MBP-A2 with CTxB, digested MBP-A2 departed from CTxB. These findings were further confirmed by Western blot (Appendix A) and MS analysis of the cell homogenate (Appendix A). MBP-A2 cleavage was cross-examined, by testing purified rCTC supplemented with protease inhibitor. The results show that MBP-A2 is degraded by proteases, and that proteolytic degradation can be avoided by adding protease inhibitors (Appendix A). The quality of the purification process is, then, fundamental for complex stability and, once formulated, rCTC can be stored at 4 °C. Furthermore, Appendix A MS data, from the intracellular fraction, shows that cleavage always occurs within the synthetic linker between MBP and the A2 helix. The proteolytic cleavage occurs because the linker sequence forms an unstructured region within the protein sequence, which is prone to degradation. This indicates that the selection of a suitable linker sequence could reduce proteolytic cleavage and, potentially, increase yields further.

A comparison, of all variants produced for 25 h and purified with the two-step purification method, is listed below in Table 2. Alongside those results are stability SEC-MALS measurements and MW weight determination, for all variants producing rCTC.

### 2.4. Analytical Investigation of rCTC Activity

rCTC was tested for its ability to bind to membrane-bound GM1, by measuring the interaction of rCTC to GM1 liposomes, with an Octet Bio-Layer Interferometry (BLI) device (Appendix A). Liposomes (100 nm), containing 10% GM1 DPPC lipid, were captured by biotinylated DSPE on an Octet streptavidin biosensor chip, and their ability to bind to rCTC was compared to liposomes without GM1 [49]. The background signal was GM1 liposomes submerged in buffer (blank/no rCTC added). The binding of rCTC increased with concentration as expected, showing that each variant produced soluble and active rCTC, but the slow dissociation of rCTC from the GM1 receptor prevented K_d_ determination. Therefore, Isothermal Titration Calorimetry (ITC) was used to compare the K_d_ for each variant binding to GM1 oligosaccharide (Figure 8 and Appendix A) [5,50]. The values of ΔH (enthalpy change) and log_10_[K_AB_] (log-binding affinity), for all rCTC samples, agreed with those for commercial CTxB (Sigma), within a 68.3% confidence interval (Appendix A). Therefore, ITC quality control concludes that all rCTC-expression conditions led to a protein that maintained uniform binding to its GM1 receptor.

### 2.5. Cell-Binding Assays for Determining the Affinity of Binding In Vitro

The GM1 receptors exposed on the surface of Jurkat cells allow the determination of rCTC binding, in a cellular environment [51]. These cells were chosen as a model cell line that expresses abundant levels of GM1 and provides information on whether the optimized yields are aligned with product activity. Flow-cytometry analysis with Jurkat cells was conducted at four different concentrations of rCTC, labeled with Alexa Fluor 647 (93 nM, 0.23 nM, 0.46 nM, 0.93 nM). Cells were incubated with rCTC AF647, for 30 min on ice, then the unbound protein was washed away, to decrease nonspecific binding. Histograms of fluorescence intensity in Figure 9 show a dose-dependent binding of the rCTC to Jurkat cells. The multivalent presentation of the GM1 glycolipid at a cell surface was expected to increase the affinity relative to the solution ITC measurements, but, remarkably, binding of the complex to cells was recorded already at a pM range (93 pM), showing higher than previously reported affinity (20 nM) [51,52,53]. To eliminate the possibility of the MBP-tag being responsible for binding, we, also, performed the binding assay in the presence of soluble maltose (Appendix A). The binding of rCTC to cells is not inhibited by maltose and remarkably, as we registered an increase in fluorescence signal, indicating enhanced binding. This might indicate that the sugar in the solution stabilizes the protein and further promotes its binding to the cell. Sugars are, often, added to increase the overall solubility and long-term stability of proteins [54,55]. However, repeating the ITC experiments in the presence of maltose led to no change in the binding interaction with GM1 oligosaccharide. Overall, these data indicate a strong binding of the complex to the receptor, which is not dependent on the MBP-tag.

### 2.6. Retrograde Transport of MBP-Tag

Characterization of the intracellular uptake of rCTC in Jurkat cells was, also, investigated (Figure 10). The CTxB protein is responsible for the internalization and retrograde trafficking of cholera toxin, through the Golgi towards its final destination, the endoplasmic reticulum [56]. Cells were imaged with confocal microscopy, to investigate the intracellular uptake of the rCTC (Figure 10). For these experiments, the concentration of fluorescently labeled rCTC was increased to 2 nM, to improve image quality. Cells were pre-incubated with rCTC AF647 for 30 min on ice, and then the unbound protein was washed away. This condition corresponded to 0 h at 37 °C in the experiment (Figure 10, top row), resulting in plasma-membrane staining by the fluorescent protein.

Incubation at 37 °C in immunofluorescence assays revealed that the rCTC is internalized and accumulates inside the cells over time, where it can be observed for up to 3 h. The MBP-tag signal is detected with primary and secondary fluorescent antibodies and shows a similar pattern after intracellular transport. The Golgi compartment was visualized utilizing a CTR433 antibody, which stains cis and medial Golgi. The fluorescence signals of rCTC AF647 and MBP-tag overlapped in this intracellular compartment, thus leading to the conclusion that rCTC is taken up by Jurkat cells, and its MBP cargo is delivered, partially, to the Golgi compartment.

## 3. Conclusions

The extracellular production in *E. coli* is a challenging task, with a plethora of variables to consider and parameters to optimize. Here, we found the solution to increase rCTC production was by selecting a weak promotor (V1 pET30a 0.09T7) and an antibiotic-resistance marker-free production strain, based on an auxotrophic marker (V1 ΔthyA pET30a). The best-performing variants were complemented with an optimized production process. A fed-batch production process was coupled with two-step purification, which allowed optimized rCTC production, at a volumetric yield of 168 mg/L. The rCTC expressed in the periplasm is subjected to proteolytic digestion, which targets the A2-linker peptide and reduces the efficiency of purification. Including those downfalls, the overall yield, still, has been improved 11-fold, due to the rational design of the construct, strains, and production process, in comparison to a host/vector combination that has been previously used (BL21(DE3) pTRBAB5-G1S). The re-engineered cholera toxin complex was fully functional, maintaining the same affinity for GM1 oligosaccharide (K_d_ ca. 40 nM). rCTC strongly recognized the full GM1 glycolipid receptor in the pM range in a model cell line (Jurkat cells) and, then, transported the MBP-tag in a retrograde fashion to the Golgi, upon cellular uptake. The rCTC reported here has potential for further modification [18] and to be exploited to deliver more diverse probes into cells in vivo, e.g., motor neurons, where it presents opportunities to provide successful tools for neuroscience.

## 4. Materials and Methods

### 4.1. Bacterial Strain

A total of four *E. coli* strains were selected for the production of rCTC and are compared to wild-type (wt) BL21 (DE3). V1 [35] and V2 [38], originating from BL21 (DE3) and BL21 AI strain, respectively, were renamed from the original publication, for simplicity purposes. BL21 (DE3) *ΔthyA* and V1 *ΔthyA* were generated by deletion of the *thyA* gene, in respect to the protocol by [57]. The *ΔthyA* selection system is a platform technology developed by enGenes Biotech that allows the production of recombinant proteins without using antibiotic resistance genes on the plasmid backbone. Plasmid-bearing cells are selected by complementing an essential gene (deleted on the genome) that is encoded on the plasmid, instead of the antibiotic-resistance marker gene.

### 4.2. Plasmids

pTRBAB5-G1S was prepared from pSAB2.1, as described in [18]. pTRBAB5-G1S (tac promoter, weak) plasmid was subcloned to pET30a-Cer plasmid (T7 promoter; Appendix A). A->C point mutation was introduced into the T7 promoter sequence to lower its strength to 0.09% (TCATACGACTCACTATAGGGGAATTGTGAGCGGATAAC). Plasmid suitable for the auxotrophic strain was generated, by replacing *kanR* with *thyA*. All materials used for cloning all constructs were purchased from New England Biolabs, Frankfurt, Germany (cloning kits) and IDT, Leuven, Belgium (primers and gBLOCKs).

### 4.3. qPCR Analysis

Selected variants for qPCR analysis were cultivated in a 100 mL volume shake flask in emi-synthetic media, as described in [41]. Cultivation was carried out at 37 °C before induction (until 1 OD) and 28 °C after induction. V1 and BL21 (DE3) variants induced with 0.5 mM Isopropyl β- D-1-thiogalactopyranoside (IPTG) (GERBU Biotechnik, Heidelberg, Germany). V1 was, also, induced with 100 mM L-arabinose (Sigma-Aldrich, A3256, Saint Louis, MO, USA). Then, 1 mg CDM was collected at 0 h, 1 h, 3 h, and 6 h post-induction, where 0 indicates the sample before induction. The RNA degradation was stopped with 5% (*v*/*v*) phenol (Sigma-Aldrich, P1307, Saint Louis, MO, USA) in EtOH stop solution (Sigma-Aldrich, 32221, Saint Louis, MO, USA). The purification was performed with a Biozym RNA purification kit, and conversion of RNA to cDNA was performed with a SuperScript™ III First-Strand Synthesis System (Thermo Fisher, 18080051, Bleiswijk, Netherlands), with a Random Hexamer Primer mixture of single-stranded random hexanucleotides, with 5′- and 3′-hydroxyl ends. The qPCR analysis was performed, as indicated in the RT-qPCR Bio-Rad protocol. The data were analyzed using CFX Manager™ Software 3.1 (Bio-Rad Laboratories, Hercules, CA, USA), with a preliminary experiment to test the efficiency of primers for cysG (housekeeping genes *E. coli* #1), rssA (housekeeping gene *E. coli* #2), gspD, gspE, gspG, mbp-a2, ctxB, slyA, and fur genes (Appendix A, IDT, Leuven, Belgium). The qPCR analysis was compared with the original strain B pTRBAB5-G1S construct (tac promotor). We used the relative normalized quantification method [58], where we analyzed the changes in gene expression at a given time point (0–6 h after induction), calculated by determining relative quantity (versus untreated control sample = ΔCq) and, then, by using normalized gene expression (versus housekeeping cysG and rssA genes = ΔΔCq). All of the graph data were set as relative to zero. The genes below 1 of Relative Normalized Expression (ΔΔCq) were classified as downregulated and above as upregulated.

### 4.4. Expression, Purification, and Offline Analysis

The Biolector^®^ small-scale cultivation, as described by [38], deviated from the original description by a lower concentration of Enzamix (*w*/*v* 0.3%) and longer pre-induction cultivation (15 h). IPTG (1 mM–0.01 mM; GERBU Biotechnik, Heidelberg, Germany) and L-arabinose (100 mM–0.25 mM; Sigma-Aldrich, A3256, Saint Louis, MO, USA) induction strategies are described in Results and Discussion. The conditions for SDS-Page analysis were carried out as in previous publications [38]. Western blot analyses (WBs) were carried out, as described in [59], replacing detection antibodies with anti-MBP (New England Biolabs, E8032, Ipswich, MA, USA). Flow-cytometry analysis, carried by Gallios flow cytometer (Beckman Coulter, Indianapolis, IN, USA), was conducted using BDTM Cell Viability Kit (Eugene, OR, USA). PI staining (often used in live-dead staining) is a widely used method to assess the viability of microbial cells. However, recent research performed by [60] showed that boosted membrane potential (ΔΨ, inside negative) results in a reversible permeability, for propidium ions in viable cells.

The batch (600 mL) and adjacent fed-batch production were carried out in the Dasgip^®^ Parallel Bioreactor Systems, Hamburg, Germany (media prepared according to Appendix A). The batch phase for all conditions reached 2.4 g CDM in 600 mL. The cultivation conditions in the feed phase were strain customized for wt BL21 (DE3), consisting of a primary long-exponential phase (µ = 0.12 h–1), lasting past induction (25 h), and a final linear phase (10.91 mL/min), calculated to 47 g/L CDM at the End of Fermentation (EoF). V strain cultivations consisted of one exponential phase (µ = 0.14 h–1), followed by 2 linear phases: (i) before induction 21.82 mL/min (4 h) and (ii) past induction 14.18 mL/min (25 h) calculated to 47 g/L CDM at EoF. Inducer concentrations were selected, based on the type of strain and condition tested. The cultivation condition was carried out at 37 °C, and production was carried out at 30 °C.

Two-step purification was performed by ÄKTA™ start (Cytiva, Uppsala, Sweden) starting with HisTrap™ FF (5 mL) (Cytiva, Uppsala, Sweden) and a self-made column with Amylose Resin High Flow (New England Biolabs, Ipswich, MA, USA; 5 mL). The binding/wash buffer for each purification was: Hepes 50 mM, 150 mM NaCl, pH 7.4. Product purified with HisTrap™ FF was eluted with binding buffer + 300 mM Imidazole (Sigma Aldrich, 56749, Vienna, Austria) and Amylose Resin with binding buffer + 20% Glucose (Sigma Aldrich, D9559, Vienna, Austria). The final product was rebuffered with Amicon^®^ Ultra Centrifugal Filters (30 kDa cutoff; Darmstadt, Germany) for Dulbecco′s phosphate-buffered saline buffer (DPBS) (PANtm Biotech, Aidenbach, Germany).

### 4.5. ELISA Quantification Assay

ELISA quantification analysis was carried out in Phosphate Buffered Saline (PBS): 10 mM Na2 HPO4, 1.8 mMKH2PO4, 137 mM, NaCl, 2.7 mM KCl, pH 7.4 with the addition of 0.05% TWEEN20, called PBS-T (Sigma Aldrich, Saint Louis, MO, USA). Blocking Buffer was *v*/*v* 20% 10× Casein Blocking Buffer (Sigma-Aldrich, B6429, Saint Louis, MO, USA) diluted with PBS-T, pH 7.4. 1.3μM GM1 ganglioside (Carbosynth, Compton, UK) methanol-coating solution was transferred by multi-channel pipette (50μL) to 96-well plates (Nunc MaxiSorp, Roskilde, Denmark). Plates were placed inside the fume hood for 1–2 h to evaporate the methanol. Then, 100 μL of blocking buffer were loaded, for each well-using dispenser, to block the plate for 1 h, 37 °C. After removing the blocking buffer, plates were washed 3x with 100 μL PBS-T buffer. Next, 50 μL of extracellular (harvested media supernatants) or cellular solubles (enzymatically lysed cell supernatants; Ref. [59] was diluted (1:2 series) and incubated for 1 h, RT. A 1:2 dilution series (0.2–0.1 mg/mL) of purified rCTC was used for concentration standards. PBS solution was used as a blank and supernatants from different than rCTC fermentation as a negative control. After 1 h incubation under RT, the plates were submitted to a second washing step. 1:10,000 dilution of the primary anti-MBP antibody (New England Biolabs, E8032, Ipswich, MA, USA) was transferred on the plate (55μL). After 1 h incubation at RT and another washing with PBS-T, secondary anti-Mouse IgG conjugated to peroxidase (Sigma-Aldrich, A2304, Saint Louis, MO, USA) was coated (50μL diluted at 1:1000 in PBST buffer), to interact with the primary antibody for 1 h under RT. The remaining part of the protocol is followed by the SIGMAFAST OPD tablet set (Sigma Aldrich, P9187, Saint Louis, MO, USA), described by Sigma-Aldrich in the product information sheets.

### 4.6. SEC-MALS, SEC, and MS Stability Analysis

Analytical Size Exclusion Chromatography (SEC) was performed, using the Äkta Purifier FPLC system (Cytiva, Uppsala, Sweden) at 4 °C. All protein samples applied were filtered through a 0.2 µm Sartorius Minisart filter (Sartorius, Goettingen, Germany). Then, 200 µL of 30 µM rCTC sample was injected onto a HiLoadTM 10/300 SuperdexTM 200 column (Cytiva, Uppsala, Sweden), and the protein was eluted with 50 mM HEPES, 150 mM NaCl, pH 7.5, at a flow rate of 1 mL/min (Sigma Aldrich, Saint Louis, MO, USA). The protein content of the elution fractions was measured via UV/vis absorbance at 280 nm.

Analytical SEC was coupled with multi-angle light scattering (MALS). The molecular weight of the constructs was analyzed by SEC, coupled with right-angle light scattering (RALS), using an OMNISEC RESOLVE/REVEAL combined system (Malvern Panalytical, Malvern, UK). Sample solutions were filtered through 0.45 μm pore size hydrophilic PVDF centrifugal filters (Millipore, Darmstadt, Germany), before analysis. Proteins were separated on a Superdex S200 increase 10/300 GL column (Cytiva, Uppsala, Sweden), maintained at 25 °C, using PBS as an isocratic mobile phase, at a flow rate of 0.5 mL/min. The autosampler chamber was maintained at 25 °C. The injection volume varied between 20 μL and 100 μL. The detectors were maintained at 25 °C. Protein concentration was measured online, by using the refractive index detector. A refractive index dn/dc of 0.185 was taken. Data were analyzed using the OMNISEC software (version 11.30).

Mass spectrometry was performed on a Bruker Daltonics maXis impactMass Spectrometer with electrospray ionization (ES)MS. The equivalent of 40 pmol rCTC was injected onto the machine, and the deconvolution of multiple charge states was analyzed using Bruker Compass Data Analysis software (version 4.3, Bruker Daltonik GmbH, Bremen, Germany).

LC-ESI-MS/MS TSQ Vantage Triple Quadrupole (Thermo Fisher Scientific, San Jose, CA, USA) analysis of peptides, originating from protease treatment, was performed by Universitat fur Bodenkultur Core Facility for Mass Spectrometry. Target MBP-A2, CTxB, and HCPs were detected in the automated search with MS/MS ion X!-Tandem (https://www.thegpm.org/tandem/, accessed on 5 July 2021). The output was analyzed against an *E. coli* database (downloaded from Uniprot https://www.uniprot.org/, accessed on 4 July 2021). The log(I) score (the base-10 log of the sum of the fragment ion intensities in the X!-Tandem MS/MS, used to make this assignment), was used as a semi-quantitative value to estimate the relative abundances of HCPs and protein of interest (MBP-A2 and CTxB).

### 4.7. Ligand Binding Assays

Preliminary activity measurements were performed by Octet^®^ RED96e 8-Channel System with Bio-Layer Interferometry assay (BLI; ForteBio, Biberach an der Riss, Germany), described by [49]. Liposomes for GM1- rCTC interaction studies were prepared with the thin-hydration method, by dissolving dipalmitoylphosphatidylcholine (DPPC; Avanti Polar Lipids, Alabaster, AL, USA) and GM1 ganglioside (*w*/*v* 10%; Carbosynth, Compton, UK) in methanol (*w*/*v* 99%; Sigma Aldrich, Saint Louis, MO, USA). Liposomes attachment to Streptavidin tips (Octet^®^ SA Biosensors) was performed with 1,2-distearoyl-sn-glycerol-3-phosphoethanolamine-N-[biotinyl(polyethyleneglycol)-2000] (DSPE; Avanti Polar Lipids, Alabaster, AL, USA). Liposomes without GM1 ganglioside were in the negative control. The homogeneity of liposomes (100 nm) was evaluated by the liposome mean diameter, and the homogeneity was measured by DLS on a Zetasizer Nano-ZS with software version 6.01 (Malvern Instruments, Malvern, UK), at 25 °C in DPBS.

ITC was performed using the MicroCal ITC200 (Malvern Panalytical, Malvern, UK), with 1 µM of the rCTC complex in the cell and approximately 50 µM GM1 pentasaccharide in the syringe (10 times the concentration of available GM1 binding sites). The ITC parameters were: 750 RPM stirring, thermosetting to 25 °C, one sacrificial injection of 0.5 µL followed by 19 injections of 2 µL each, 120 s between injections, with injections lasting 4 s each. Each sample was utilized for two titrations, to enable global analysis, which was performed using NITPIC (version1.2.2, Brautigam–UT Southwestern, Dallas, TX, USA), SEDPHAT (version 14.0, National Institutes of Health, Bethesda, MD, USA) and GUSSI (version 1.4.1, Brautigam–UT Southwestern, Dallas, TX, USA) software. The error analysis was conducted using ‘Generate 1-dimensional error surface projection’. Error analysis was done (independently) for the Log_10_[K_AB_] and ΔH parameters by alternating between Simplex and Marquardt–Levenberg fitting algorithms until converged.

### 4.8. Cell-Based Binding Assays

The reagents were obtained from commercial sources: RPMI 1640, PBS (−/−), FCS, and L-glutamine were purchased from Gibco (Thermo Fisher Scientific Inc., Rockford, IL, USA). The human T lymphocyte Jurkat cell line (American Type Culture Collection, TIB-152, Thermo Fisher Scientific Inc., Rockford, IL, USA) was cultured in Roswell Park Memorial Institute (RPMI) medium, supplemented with 10% fetal calf serum (FCS), 2 mM L-glutamine, 5 μg/mL Penicillin and Streptomycin, 1% HEPES, and 50 μM 2-mercaptoethanol, under sterile conditions at 37 °C and 5% CO2. Cells were stimulated with indicated final concentrations of rCTC in the medium, for the indicated time points.

Alexa Fluor 647 dye (Thermo Fisher Scientific Inc., Rockford, IL, USA) mono-reactive NHS ester was used to label rCTC, in accordance with the instructions of the manufacturer. Fluorescent dye was dissolved at a final concentration of 1 mg/mL in water-free DMSO (Carl Roth GmbH & Co. KG, Karlsruhe, Germany), aliquoted, and stored at −20 °C before usage. For the labeling reaction, 100 µL of protein (3 mg/mL) was supplemented with 10 µL of a 1 M NaHCO3 (pH 8.5) solution. The molar ratio between dye and protein was set to 6:1, and the final ratio resulted in 4:1. The labeling mixture was incubated at 4 °C for 90 min under continuous stirring, and uncoupled dyes were separated using Zeba Spin desalting columns (7 k MWCO, 0.5 mL, Thermo Fisher Scientific Inc., Rockford, IL, USA). Labeled protein was stored at 4 °C, protected from light.

For flow cytometry analysis, 1 × 10^5^ cells were counted and transferred to a U-bottom 96-well plate (Sarstedt AG & Co. KG, Numbrecht, Germany). To quantify protein binding to cell-surface receptors, cells were incubated with increasing concentrations (93 pM, 234 pM, 0.46 nM, and 0.93 nM) of fluorescently labeled rCTC, for 30 min at 4 °C and protected from light, compared to PBS-treated cells as the negative control. Subsequently, cells were centrifuged at 1600× *g* for 3 min at 4 °C and washed twice with FACS buffer (PBS (−/−) supplemented with 3% FCS *v*/*v*). After the last washing step, the cells were re-suspended with FACS buffer and transferred to FACS tubes (Kisker Biotech GmbH Co. KG, Steinfurt, Germany) on ice and protected from light. The fluorescence intensity of treated cells was monitored at FACS Gallios (Beckman Coulter Inc., Brea, CA, USA) and further analyzed using FlowJo V.10.5.3.

### 4.9. Immunofluorescence and Confocal Microscopy

Between 5 and 6 × 10^4^ Jurkat cells were transferred to a U-bottom 96-well plate and incubated with 2 nM of fluorescently labeled rCTC diluted in the medium for 30 min at 4 °C, in absence of light. At the end of pre-incubation, cells were centrifuged at 1600× *g* for 3 min at 4 °C and washed once with PBS, to remove the excess protein in the solution. Cells were then resuspended in a fresh medium and incubated at 37 °C for the indicated time points. Subsequently, cells were fixed with 4% paraformaldehyde for 15 min at RT and quenched with 50 mM NH_4_Cl for 10 min at RT. The membrane was permeabilized, and cells were blocked by 0.2% Saponin in 3% BSA in PBS (*w*/*v*) for 30 min. Samples were incubated with 100 µL target primary antibodies (1:100) for 1 h at RT. After three washes with PBS (−/−), cells were stained with 100 µL fluorescently labeled secondary antibodies (1:200) for 30 min at RT in the dark. Nuclei were counterstained with DAPI (5 × 10^−9^ g/L), and the cells were resuspended in 20 µL of 20% glycerol in PBS. Samples were imaged through a laser scanning confocal microscope system (Nikon Eclipse Ti-E inverted microscope equipped with Nikon A1R confocal laser scanning system, 60× oil immersion objective, numerical aperture (NA) of 1.49, with four lasers: 405 nm, 488 nm, 561 nm, and 640 nm). The images were further analyzed using NIS-Element Confocal 4.20 from Nikon and ImageJ 1.52a from Laboratory for Optical and Computational Instrumentation.

The following antibodies were obtained from commercial sources: Maltose Binding Protein/MBP Rabbit Polyclonal Antibody (Thermo Fisher Scientific Inc., Rockford, IL, USA, Cat. No. BS-2967R), Goat Anti-Rabbit DyLight488 (Thermo Scientific/Pierce, Waltham, MA, USA, Cat. No. #35552), and Goat Anti-Mouse Alexa546 (Jackson Immunoresearch, West Baltimore Pike. West Grove, PA, USA, Cat. No. A11030). The medial Golgi was detected by a custom-made mouse CTR433 antibody (Institute Curie, Paris, France). RPMI 1640, PBS (−/−), FCS, and L-Glutamine were all purchased from Gibco (Thermo Fisher Scientific Inc., Waltham, MA, USA). The following chemicals were obtained from Roth (Carl Roth GmbH & Co. KG, Karlsruhe, Germany): BSA, DAPI, glycerol, NH_4_Cl, paraformaldehyde, and sodium hydrogen carbonate. Saponin was obtained from Sigma-Aldrich (Sigma-Aldrich Chemie GmbH, Darmstadt, Germany).

## Figures and Tables

**Figure 1 toxins-14-00396-f001:**
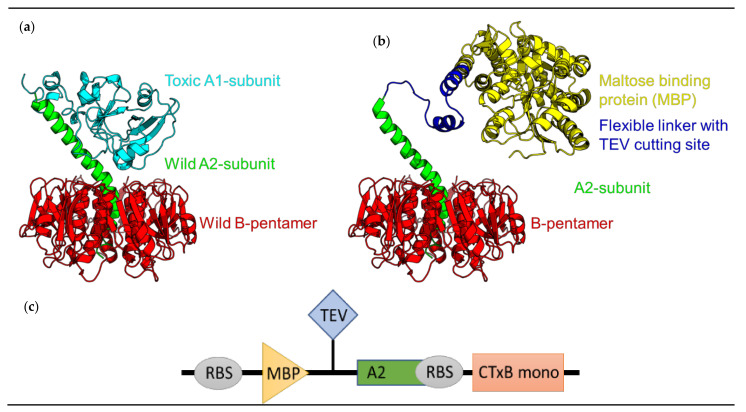
Composition of re-engineered cholera toxin complex (rCTC) and its soluble MBP tag. (**a**) Native CTx and (**b**) Toxic A1 subunit are replaced with MBP-tag, to enable purification of A2 helix with Amylose Affinity Chromatography (AAC). Between MBP-tag and A2 helix, there is a flexible linker with a cleavable site. CTxB is captured, by natively occurring histidine, purifiable by Ni+ affinity chromatography, also referred to as Immobilise Metal Affinity Chromatography (IMAC). MBP-tag can be cleaved post-purification, with TEV protease. (**c**) Expression-unit design for polycistronic expression of rCTC. MBP-A2 is expressed as one and separate from CTxB, which is expressed as a monomer that folds into a pentamer. MBP-A2 and CTxB pentamer interact to form rCTC.

**Figure 2 toxins-14-00396-f002:**
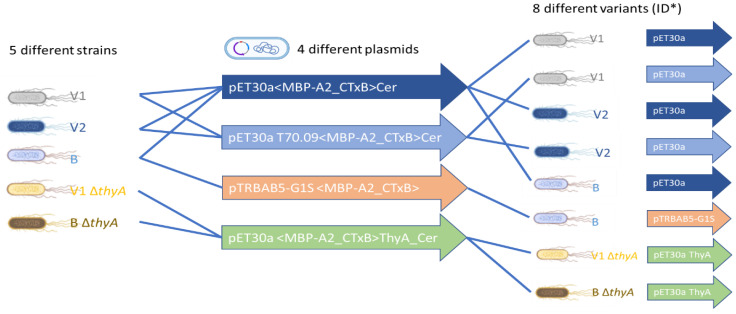
Combination of all host–vector combinations. Five different strains: (i) V1, (ii) V2, (iii) B, (iv) V1 *ΔthyA*, and (v) B *ΔthyA* were matched with four different plasmids: (1) pET30a <MBP-A2_CTxB>Cer, (2) pET30a 0.09T7 <MBP-A2_CTxB>Cer, (3) pTRBAB5-G1S, and (4) pET30a <MBP-A2_CTxB>ThyA_Cer, leaving eight variants as a result. Created with BioRender.com. * Table 1.

**Figure 3 toxins-14-00396-f003:**
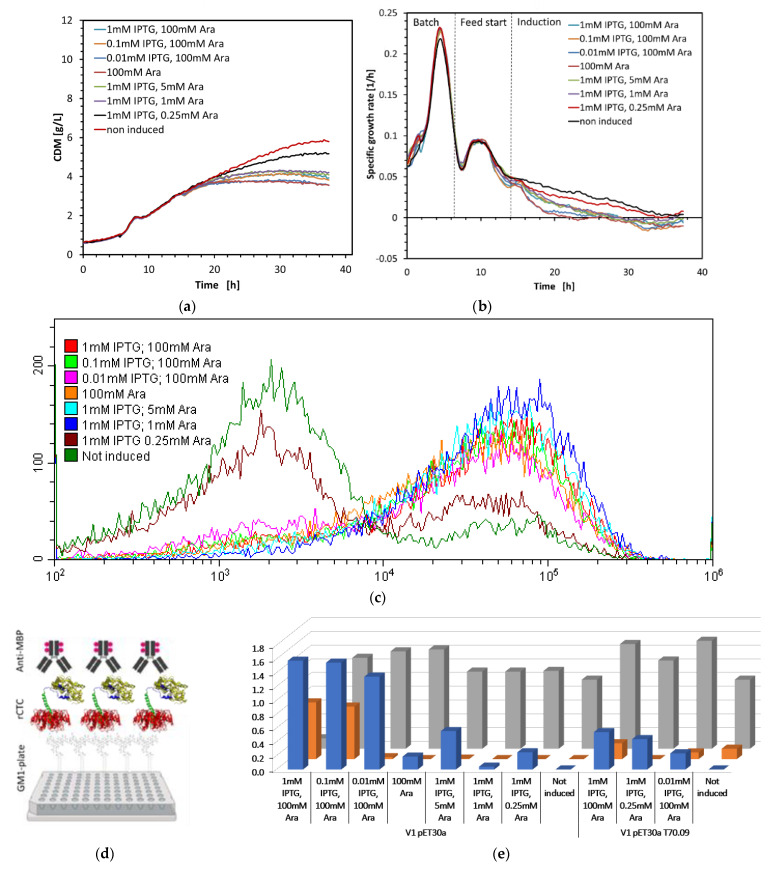
Biolector^®^ cultivation and analysis of rCTC, produced by V1 pET30a variant. (**a**) CDM [g/L] was obtained in real-time corresponding to (**b**) growth rate with indicated process course at different IPTG and Ara concentrations. (**c**) The flow-cytometry analysis, with PI staining. (**d**) ELISA design, in which the 96-well plate is coated with ganglioside GM1, to allow capture of rCTC through its CTxB subunit and detection via an anti-MBP antibody. Created with BioRender.com. (**e**) ELISA determined specific extracellular [mg/L] and intracellular [mg/g] rCTC product concentration and corresponding dsDNA [fold-non-induced].

**Figure 4 toxins-14-00396-f004:**
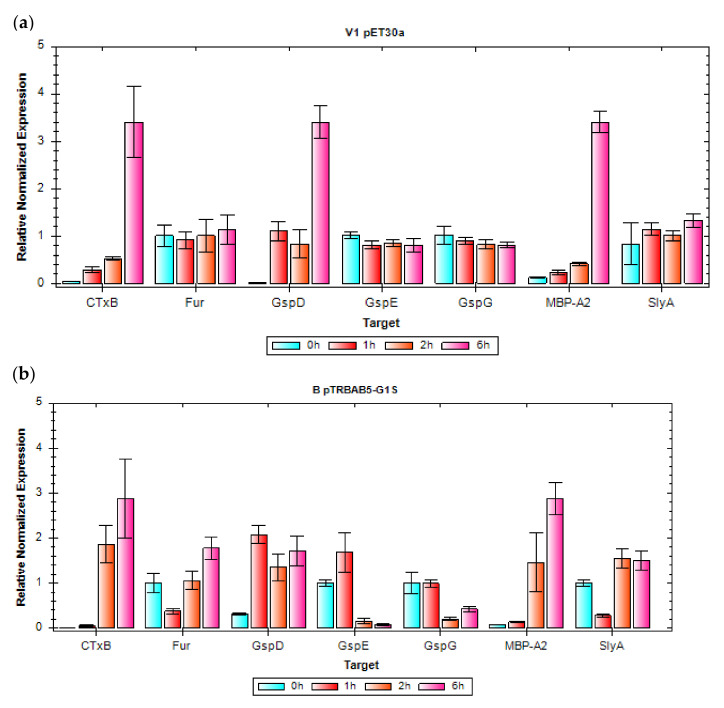
Relative normalized expression (ΔΔCq) of *ctxB*, *fur*, *gspD*, *gspE*, *gspG*, *mbp-a2*, and *slyA* from rCTC. (**a**) V1 pET30a (**b**) B pTRBAB5-G1S expression was analyzed against reference *cysG* and *rssA* housekeeping genes.

**Figure 5 toxins-14-00396-f005:**
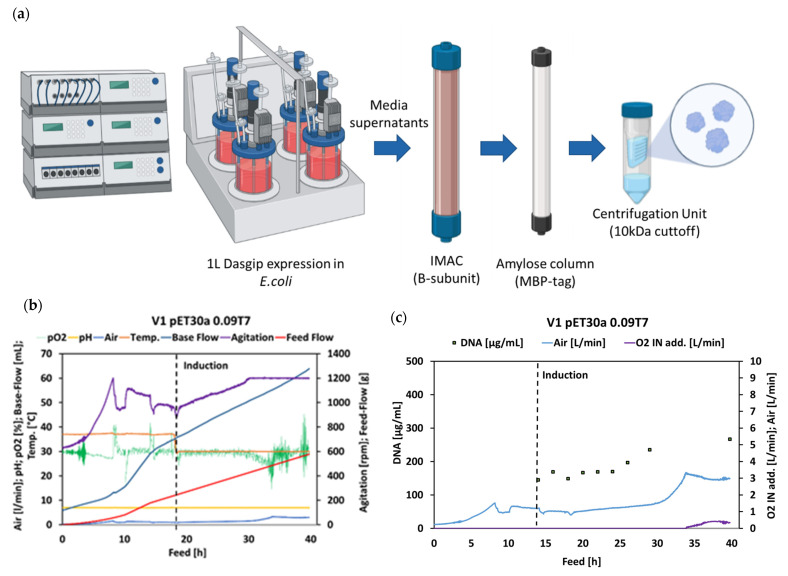
Production process parameters and rCTC concentrations. (**a**) rCTC production (0.5 mM IPTG 100 mM Ara induction for V strains; 0.5 mM IPTG for B strain), purification (HisTrap™ FF and Amylose Resin High Flow), and formulation process (Amicon^®^ Ultra Centrifugal Filters). Created with BioRender.com. (**b**) Process diagram with recorder pO2, pH, Air, Temperature (Temp), Base Flow, Agitation, and Feed Flow parameters. (**c**) Process gas demands and extracellular DNA yields (cell integrity indicator). (**d**) Final yields of extracellular rCTC purified and formulated, as described in a.

**Figure 6 toxins-14-00396-f006:**
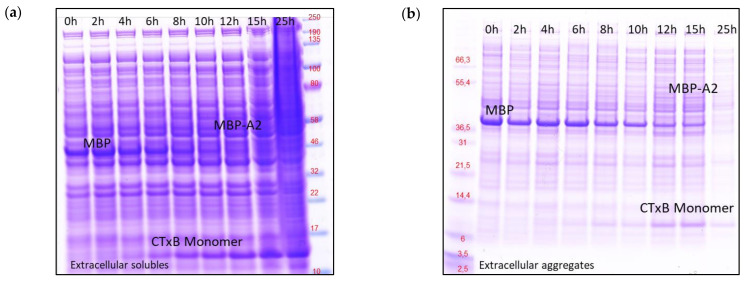
SDS-PAGE gels analysis of V1 pET30a T70.09. (**a**) Extracellular soluble and (**b**) aggregates of media supernatants were collected between 0 h–25 h, after induction. The gels show MBP, MBP-A2, and CTxB (monomer), detected under reducing conditions.

**Figure 7 toxins-14-00396-f007:**
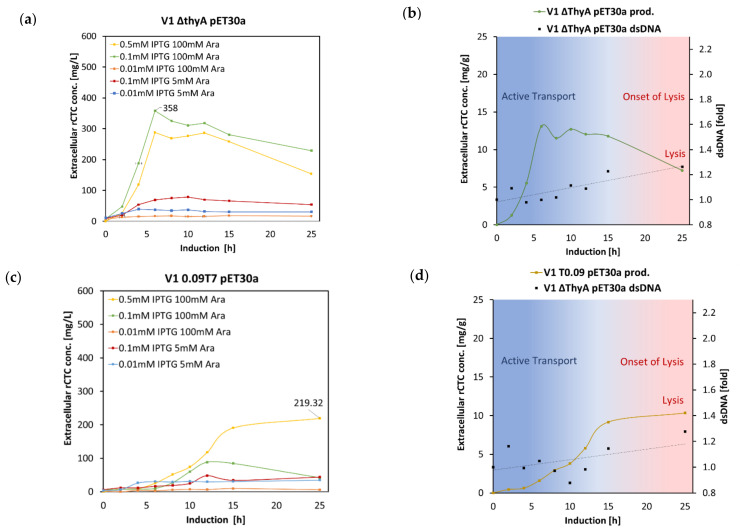
ELISA-determined rCTC yields and comparison of different IPTG and arabinose concentrations. (**a**) V1 ΔthyA pET30a induced at different condition and corresponding diagram (**b**) for extracellular rCTC concentration [mg/L] versus dsDNA [fold]. (**c**) V1 pET30a T70.09 induced at a different condition and corresponding diagram (**d**) for extracellular rCTC concentration [mg/g] versus dsDNA [fold]. Active transport of rCTC slowly transits towards passive transport, as the dsDNA complements rCTC increase after 10 h.

**Figure 8 toxins-14-00396-f008:**
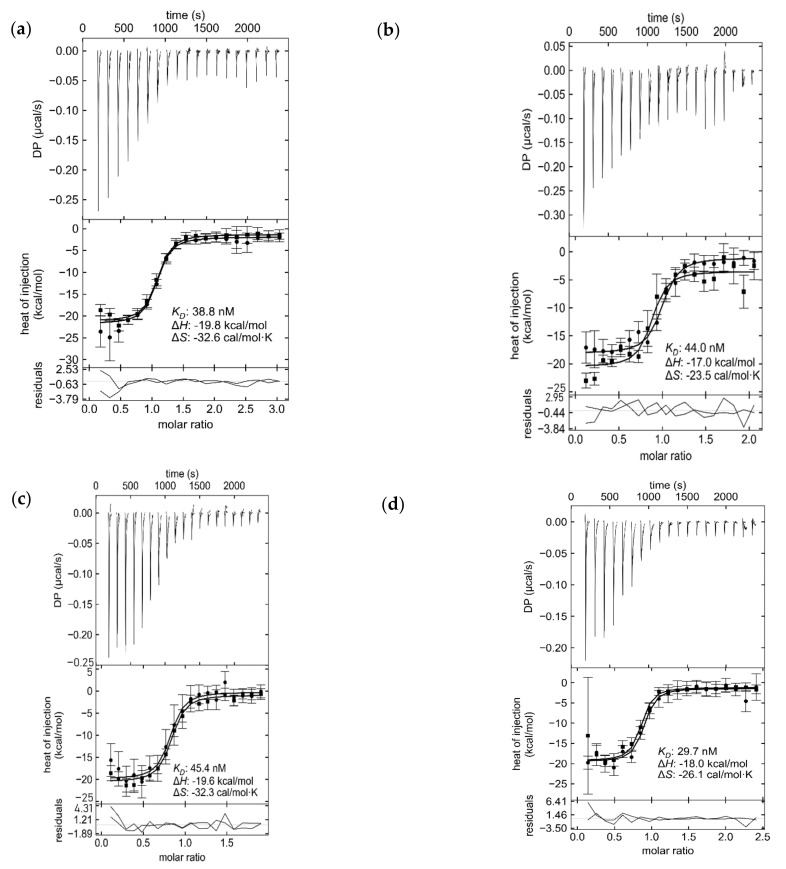
ITC analysis of formulated rCTC, produced by (**a**) SIGMA, C9903-2MG, (**b**) V1 pET30a T70.09, (**c**) V1 ΔthyA pET30a, and (**d**) B pTRBAB5-G1S.

**Figure 9 toxins-14-00396-f009:**
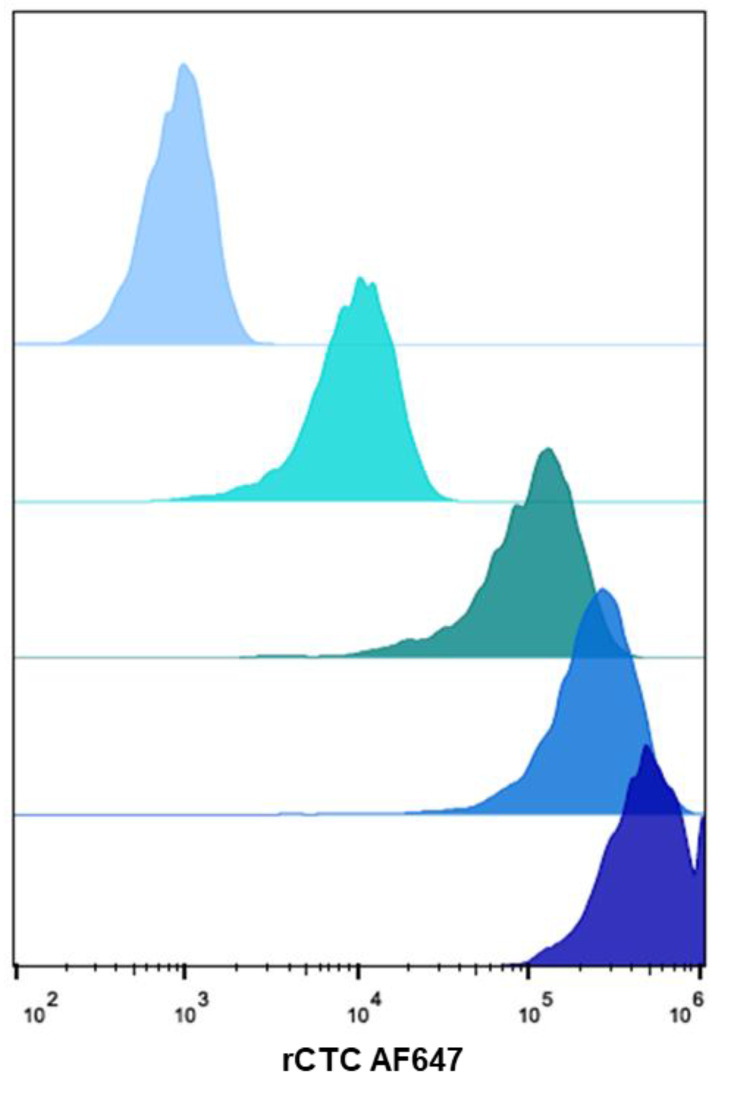
Flow-cytometry analysis of gated living Jurkat cells, incubated with increasing concentrations of fluorescently labeled rCTC AF647, for 30 min at 4 °C. Histograms of fluorescence intensity reveal a dose-dependent binding of the protein complex to the cell surface. From top to bottom: negative control, 93 pM, 0.23 nM, 0.46 nM, and 0.93 nM.

**Figure 10 toxins-14-00396-f010:**
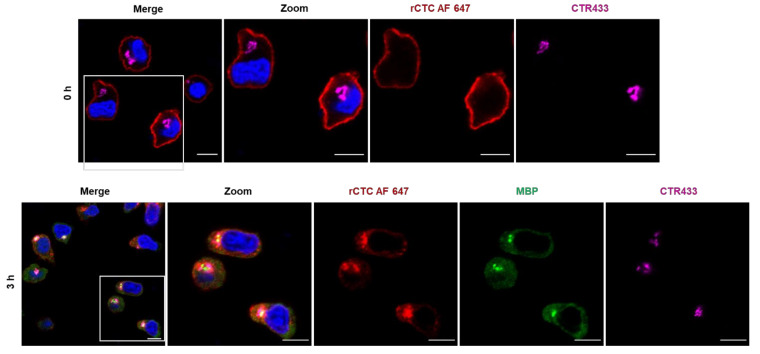
Fluorescence imaging revealed intracellular uptake of rCTC on Jurkat cells. Confocal imaging of fluorescently labeled rCTC (red), incubated with Jurkat cells at different time points (top row: time = 0 h; bottom row: time = 3 h). rCTC AF 647 is internalized, and the MBP-tag (green) overlays with the complex at the Golgi compartment (magenta). Nuclei were counterstained by DAPI (blue). Scale bars: 10 μm.

**Table 1 toxins-14-00396-t001:** List of all variants producing rCTC.

ID	Full Name	Original Strain Name	Plasmid	Resistance Marker	Promotor
**V1 pET30a**	V1 pET30a <MBP-A2_CTxB>Cer	BL21(DE3)::TN7 <pAraB-gp2>ΔaraBADC	pET30a<>Cer	Kanamycin	T7
**V1 0.09T7 pET30a**	V1 pET30a T70.09 <MBP-A2_CTxB>Cer	BL21(DE3)::TN7 <pAraB-gp2>ΔaraBADC	pET30a<>Cer	Kanamycin	T7 0.09%
**V2 pET30a**	V2 pET30a <MBP-A2_CTxB>Cer	BL21-AI<gp2>	pET30a<>Cer	Kanamycin	T7
**V2 0.09T7 pET30a**	V2 pET30a T70.09 <MBP-A2_CTxB>Cer	BL21-AI<gp2>	pET30a<>Cer	Kanamycin	T7 0.09%
**V1** ** *ΔthyA* ** **pET30a**	V1 *ΔthyA* pET30a <MBP-A2_CTxB>ThyA_Cer	BL21(DE3)::TN7 < pAraB-gp2>ΔaraBADC *ΔthyA*	pET30a<>ThyA_Cer	-	T7
**B** ** *ΔthyA* ** **pET30a**	BL21 (DE3) *ΔThyA* pET30a <MBP-A2_CTxB>ThyA_Cer	BL21(DE3) *ΔthyA*	pET30a<>ThyA_Cer	-	T7
**B pET30a**	BL21 (DE3) pET30a <MBP-A2_CTxB>Cer	BL21 (DE3)	pET30a<>Cer	Kanamycin	T7
**B pTRBAB5-G1S**	BL21 (DE3) pTRBAB5-G1S <MBP-A2_CTxB>	BL21(DE3)	pTRBAB5-G1S	Ampicillin	Tac

**Table 2 toxins-14-00396-t002:** Comparison of rCTC productivity and stability of the complex in purified samples and MW.

ID	Formulated rCTC Conc. [mg/L]	Formulated rCTC Yield Per CDM [mg/g CDM]	rCTC Detected by SEC-MALS [%]	MW Determined by LC-SEC [kDa] *
**V1 pET30a**	79	1.9	29.8	106.6
**V1 pET30a T70.09**	168	6.9	33.2	106.9
**V2 pET30a**	63	1.7	23.2	106.6
**V2 pET30a T70.09**	18	0.4	20.8	105.0
**V1 ΔthyA pET30a**	141	5.8	15.4	106.0
**B ΔthyA pET30a**	32	0.9	20.2	106.7
**B pET30a**	68	2.5	21.4	108.0
**B pTRBAB5-G1S**	15	0.4	79.0	107.3

* The error of the method was set to be 10%.

## Data Availability

The datasets generated and/or analyzed during the current study are available from the corresponding authors, on reasonable request.

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
