# Peer review of "In-Depth Characterization of a Re-Engineered Cholera Toxin Manufacturing Process Using Growth-Decoupled Production in Escherichia coli"

_toxins, 2022, doi:10.3390/toxins14060396_

Round 1

Reviewer 1 Report

This manuscript is trying to solve a problem of production of Ctx. Their methodology is very good and quality of data and presentation is acceptable. However, I have some questions.

  1. Figure 3e: proper labelling is required in x - axis.
  2. Line 231, repeated sentence.
  3. Please provide the reason for not including Hns gene in qPCR reaction.
  4. Over production of SlyA enhances hlyE expression and that antagonize the negative effects of Hns genes. Shouldn't his be tested as well?
  5. Authors concluded T2SS is active in V1 strain. But no further experiment on this? I think they should confirm this pathway. T2SS is of two subtypes: T2SSa and T2SSb, which one is dominant?
  6.  Another thing to explore is whether it is Tat or Sec pathway.
  7. Any reason for why T2SS is not active in B strain?
  8. Don't see CtX A subunit?
  9. I am not sure that protein amount is for only Ctx or its with tags and impurities. Please make it clear.
  10. Some figures from supplementary figures should be in the main article, for example Western Blot.

Author Response

Thank you for contributing to this manuscript. The comments from your revision have been implemented as it is indicated in the attached documents.

 Point-by-point response to the reviewer’s comments:

  1. Figure 3e: proper labelling is required in x - axis.

The Figure 3e was changed with visible and clear x-axis labelling.

  1. Line 231, repeated sentence.

Line 231: repeated sentence deleted.

  1. Please provide the reason for not including Hns gene in qPCR reaction.

Line 240-242: Sentence was added: “DNA-binding transcriptional dual regulator hns was excluded since it inhibits transcription initiation of gspD the same as fur.”

  1. Over production of SlyA enhances hlyE expression and that antagonize the negative effects of Hns genes. Shouldn't his be tested as well?
  2. Authors concluded T2SS is active in V1 strain. But no further experiment on this? I think they should confirm this pathway. T2SS is of two subtypes: T2SSa and T2SSb, which one is dominant?
  3. Another thing to explore is whether it is Tat or Sec pathway.

Line 252-254: “In contrary to B strain, no upregulation for slyA and fur was observed in V1 strain. Further testing is still required for determining the role of those genes in rCTC expression.”- with this sentence we wish to inform the reader that further investigation will be required.

In this section, we intended to show how one can examine chances of extracellular expression in a particular E. coli strain. Complete analysis of T2SS pathway, its secretory mechanism dedicated solely to rCTC was not our intention in this manuscript. We wish to follow up on this research in next publication along with full transcriptomic data on both strains. Thank you for understanding.

  1. Any reason for why T2SS is not active in B strain?

T2SS (gspD) is active in B and V1 strains (Reference 48 and Figure 4). The main difference is that gspD is upregulated more and later in the production of rCTC.

  1. Don't see CtX A subunit?

 There is no A1 subunit in our rCTC construct, only A2 non-toxic part as described in subsection 2.1 and shown in figure 1b.

  1. I am not sure that protein amount is for only Ctx or its with tags and impurities. Please make it clear.

 In line 98-100 and Figure 2a we describe that MBP-tag is not cleaved off in this process. In table 2 and supplementary S5, 8 and 9 we describe all impurities detected during purification process and storage.

  1. Some figures from supplementary figures should be in the main article, for example Western Blot. We thank you for this suggestion, but we believe that the main article already contains a significant number of figures. We have therefore decided to put auxiliary data in the supplementary to keep the focus on production optimization, key quality control (ITC) and ability of rCTC complex to deliver MBP intracellularly.

Reviewer 2 Report

In this manuscript, the study presented an optimized process for rCTC production and showed an 11-fold increase compared to previously used BL21(DE3) pTRBAB5-G1S. More specifically, the authors designed the rCTC construct, optimized the fermentation process, and measured rCTC stability and quality. Overall, the experiment’s design is reasonable, and there are sufficient data to convince me. Also, the results were clearly presented. Therefore, I have only a few comments below.

Minor comments:

Line 16, a host/vector combination that has been previously used (BL21(DE3) pTRBAB5-G1S). Add references.

Line 231-232, sentences repeated.

Line 258-260, “we detected higher product titers present in the extracellular space in comparison to the yields observed following small-scale fed-batch-like cultivation in BioLector.” Add Fig.3 and Fig.5 to the end of the sentence as data support.

In Fig.3, the rCTC yield in V1 pET30a > V1 pET30a T0.09, Fig. 5 the yield in V1 pET30a < V1 pET30a T0.09. Why is the yield reversed when formulation volume magnifies from small-scale to 1L scale?

According to Fig.5, is it possible that V1ΔthyA pET30a T0.09 will further increase extracellular rCTC concentration?

Line 295, “which is consistent with our qPCR data (Figure 3a)”. Fig.3a is not qPCR results.

In Fig.7a, 0.5mM IPTG 100mM Ara and 0.1mM IPTG 100mM Ara labeled with the same color. This panel showed that it was 0.5 mM IPTG / 100 mM Ara induction condition that produced 358mg/L rCTC.

Line 342-344, “V1 Δ thyA pET30a gives the highest levels of expression, but the best-performing strain after 25h is V1 pET30a T70.09 at the highest induction strength (0.5 mM IPTG /100 mM Ara) with a long production time that gives high yields of stable rCTC.” Why did you choose the 25h time point as a standard to harvest the product? If you gather between 5h-15h in V1 Δ thyA pET30a, you can get a higher extracellular rCTC concentration.  

Line 344-346, “V1 pET30a T70.09 delivers holotoxin at a lower rate into periplasm than V1 Δ thyA pET30a which can affect the T2SS secretory machinery (GspD OM protein)”. It can not explain the higher extracellular yield in V1 Δ thyA pET30a than in V1 pET30a T70.09.

Author Response

Thank you for contributing to this manuscript. The comments from your revision have been implemented as it is indicated in the attached documents.

 Point-by-point response to the reviewer’s comments:

Line 16, a host/vector combination that has been previously used (BL21(DE3) pTRBAB5-G1S).

We would support this comment; however, it is unlikely the editor would keep the reference in abstract. It will not be carried forward in any website that lists the paper. The reference is the text in line 132 and at line 105 and many more as Reference 18.

Line 231-232, sentences deleted.

Line 258-260, “we detected higher product titers present in the extracellular space in comparison to the yields observed following small-scale fed-batch-like cultivation in BioLector.”  Fig.3 and Fig.5 added to the end of the sentence as data support.

In Fig.3, the rCTC yield in V1 pET30a > V1 pET30a T0.09, Fig. 5 the yield in V1 pET30a < V1 pET30a T0.09. Why is the yield reversed when formulation volume magnifies from small-scale to 1L scale?

Small-scale Biolector fed-batch-like fermentation have often different results then real fed-batch (Dasgip 1L). We used this small-scale device to select the optimal induction strength for each variant.

The final yields at 1L are V1 pET30a < V1 pET30a T0.09 because all cultivation was induced and harvested after 25h. This means we observe the same effect as shown in Figure 7a and b. Variant V1 pET30a with full strength promotor reach maximal yield of rCTC earlier then V1 pET30a T0.09 and after a while start to lyse. In Small-scale Biolector fed-batch like cultivation, we observe an opposite rCTC yields in V1 pET30a > V1 pET30a due to different feed rate and small cell density. In future, we wish to use new generation Biolector with fluidic system allowing for real fed-batch cultivation in small scale.

According to Fig.5, is it possible that V1ΔthyA pET30a T0.09 will further increase extracellular rCTC concentration?

We believe this to be the case, but it requires further re-cloning of the construct and several cultivations. We also want to test V1ΔthyA pET30a in chemostat type cultivation and include those experiments in our next publication.

Line 350-351 added: “Benefits of combining two variants into V1 Δ thyA pET30a T70.09 are yet to be researched.”

Line 295, “which is consistent with our qPCR data (Figure 3a)”. Fig.3a changed to Figure 4a.

In Fig.7a, 0.5mM IPTG 100mM Ara and 0.1mM IPTG 100mM Ara labeled with the same color. This panel showed that it was 0.5 mM IPTG / 100 mM Ara induction condition that produced 358mg/L rCTC.

Figure 7a was corrected.

Line 342-344, “V1 Δ thyA pET30a gives the highest levels of expression, but the best-performing strain after 25h is V1 pET30a T70.09 at the highest induction strength (0.5 mM IPTG /100 mM Ara) with a long production time that gives high yields of stable rCTC.” Why did you choose the 25h time point as a standard to harvest the product? If you gather between 5h-15h in V1 Δ thyA pET30a, you can get a higher extracellular rCTC concentration. 

Line 347-348: The sentence was change to: “V1 Δ thyA pET30a gives the highest levels of expression with optimal harvest between 5-15h after induction.”

Line 344-346, “V1 pET30a T70.09 delivers holotoxin at a lower rate into periplasm than V1 Δ thyA pET30a which can affect the T2SS secretory machinery (GspD OM protein)”. It cannot explain the higher extracellular yield in V1 Δ thyA pET30a than in V1 pET30a T70.09.

Line 352-358 was changed to: “Overall, the delivery of holotoxin at a higher rate into periplasm can affect the T2SS secretory machinery (GspD OM protein) (26). Our data from the shake flask experiment using qPCR analysis was further supported by transcriptomics, as GspD is upregulated at 1 h and 6 h after induction for empty V1 strain (Figure S4). The GspD expression increases even further for a strain bearing the pET30a<MBP-A2 CTxB>Cer plasmid and imposes greater demands on OM protein ex-pression machinery leading to cell lysis”.

Our objective we to show the reader that limitation of extracellular expression points out to be GspD driven.

Reviewer 3 Report

An interesting and well written manuscript, dedicated to producing a recombinant cholera toxin complex tracer.

I only have some minor remarks, in order to to clarify a few fine details of your work.

R1. Please consider that reference [35], related to the ColE1 resolution fragment, is actually:  ... ACS Synth. Biol. 2020, 9, 1336−1348. https://dx.doi.org/10.1021/acssynbio.0c00028

R2. Some of the yields reported in Figure 5.d seems to be quite large, considering the complex procedure described in section 4.4. Were those values recalculated in scaled conditions, or are reported as raw values?

R3. In row 613, please indicate the details of the software applications you have used. As an example: NITPIC (Brautigam - UT Southwestern, Dallas, Texas).

R4. Regarding your mention in row 614, have you worked using SEDFIT (the older version) or SEDPHAT? There are significant differences in error surface projection algorithm of them. In the one in SEDFIT, no corrections are made for standard thermodynamic parameters. 

Author Response

Thank you for contributing to this manuscript. The comments from your revision have been implemented as it is indicated in the attached documents.

Point-by-point response to the reviewer’s comments:

R1. Please consider that reference [35], related to the ColE1 resolution fragment, is actually:  ... ACS Synth. Biol. 2020, 9, 1336−1348. https://dx.doi.org/10.1021/acssynbio.0c00028

Correction of reference implemented. Line 780-781.

R2. Some of the yields reported in Figure 5.d seems to be quite large, considering the complex procedure described in section 4.4. Were those values recalculated in scaled conditions, or are reported as raw values?

Those values are recalculated in scale condition (1L), as described in figure 5a. There is huge difference between small-scale fed-batch-like cultivation (Biolector) and large scale 1L (Dasgip) due to different feed process and cell density in culture. We use small scale for selecting inducer concentration and process parameters and larger scale for actual production.

R3. In row 613, please indicate the details of the software applications you have used. As an example: NITPIC (Brautigam - UT Southwestern, Dallas, Texas).

Correction in line 612-614 to: “The liposome means diameter and the homogeneity were measured by DLS on a Zetasizer Nano-ZS with software version 6.01 (Malvern Instruments, Malvern, UK) at 25 °C in DPBS.” More detailed information about the method can be found in Reference 49.

R4. Regarding your mention in row 614, have you worked using SEDFIT (the older version) or SEDPHAT? There are significant differences in error surface projection algorithm of them. In the one in SEDFIT, no corrections are made for standard thermodynamic parameters. 

Line 623-625 change to: “NITPIC (V.1.2.2, Brautigam – UT Southwestern, Dallas, Texas, USA), SEDPHAT (V.14.0, National Institutes of Health, Bethesda, Maryland, USA), and GUSSI (V.1.4.1, Brautigam – UT Southwestern, Dallas, Texas, USA) software.”

Reviewer 4 Report

This is a manuscript on a system applied to cholera toxin transport into cells, in which various conditions are examined in detail to optimize the amount of production. The manuscript is a good paper with very detailed description of the expression conditions and functional confirmation of the produced protein. The manuscript shows great promise for applications in cell biology and vaccine development.

+MBP is used as delivering cargo, but it would be better to use labeling such as GFP for direct observation.

+From the data in Figure 5, comparing V1 pET30a and V1 dthyA pET30a, the latter has a higher expression, and if so, is V1 dthyA pET30a T70.09 much higher expressed? Why is this combination not examined?

+If various proteins can be injected into cells instead of MBP, it will be very useful for the development of cell biology.

P.7, L229-232: duplicate. 'and negatively regulated by Fur and Hns, ...SlyA.'

p.12, L354: Table 3 is nowhere to be found. (Table 3) -> (Table 2)

p.13, L371: listed below in Table 1. -> listed below in Table 2.

p.19, L570-L572: I don't know the difference between MALS and RALS.

p.22, L725: ref #16 ofMicrobiological -> Please describe it correctly.

p.22, L739: ref #22 Res Sq. -> Please list the volume, issue, and pages.

p.23, L767 & L781: ref #35 & #42 -> I believe they are identical.

There are miscellaneous deficiencies in the References. Please check them out.

Author Response

Thank you for contributing to this manuscript. The comments from your revision have been implemented as it is indicated in the attached documents.

 Point-by-point response to the reviewer’s comments:

+MBP is used as delivering cargo, but it would be better to use labelling such as GFP for direct observation.

This would indeed be correct for application like confocal microscopy. However, in our publication motives of using MBP-tag instead are described in subsection 2.1 or Figure 5a. In there, we show to reader benefit of MBP-tag to capture complete rCTC complex with two step purification (His-trap and Amylose column).

+From the data in Figure 5, comparing V1 pET30a and V1 dthyA pET30a, the latter has a higher expression, and if so, is V1 dthyA pET30a T70.09 much higher expressed? Why is this combination not examined?

We believe this to be the case, but it requires further re-cloning of the construct and several cultivations. We also want to test V1ΔthyA pET30a in chemostat type cultivation and include those experiments in our next publication.

Line 350-351 added: “Benefits of combining two variants into V1 Δ thyA pET30a T70.09 are yet to be researched.”

+If various proteins can be injected into cells instead of MBP, it will be very useful for the development of cell biology.

In future research we intend to replace MBP-tag or modify it to either introduce fluorophore or a drug into the cells.

P.7, L229-232: duplicate. 'and negatively regulated by Fur and Hns, ...SlyA.'

The duplication was deleted.

p.12, L354: Table 3 is nowhere to be found. (Table 3) -> (Table 2)

The table number was changed to 2.

p.13, L371: listed below in Table 1. -> listed below in Table 2.

The table number was changed to 2.

p.19, L570-L572: I don't know the difference between MALS and RALS.

Line 578 sentence changed to: “Analytical SEC was coupled with multi-angle light scattering (MALS).”

Multi angle light scattering (MALS) versus right angle light scattering (RALS).

p.22, L725: ref #16 ofMicrobiological -> Please describe it correctly.

Line 737: the reference was corrected.

p.22, L739: ref #22 Res Sq. -> Please list the volume, issue, and pages.

Reference 22 was corrected.

p.23, L767 & L781: ref #35 & #42 -> I believe they are identical.

Reference 42 was deleted.

There are miscellaneous deficiencies in the References. Please check them out.

Correction was implemented.

Round 2

Reviewer 1 Report

Line 240: Space between "as fur".